# Determinants of the Cardiovascular Capacity of Amateur Long-Distance Skiers during the Transition Period

**DOI:** 10.3390/diagnostics10090675

**Published:** 2020-09-05

**Authors:** Natalia Grzebisz

**Affiliations:** Faculty of Dietetics, Vistula School of Hospitality, 02-787 Warsaw, Poland; n.grzebisz@gmail.com

**Keywords:** biomarkers, amateur, sports cardiology

## Abstract

The aim of this study was to identify determinants of the cardiovascular capacity of 16 male amateur long-distance skiers during the transition period. These factors can vary from amateur marathon skiers, who represent a sort of midpoint between inactive people and professional athletes. Cardiovascular capacity depends mainly on the volume and intensity of the training, which are different between these groups. Finding the factors affecting heart condition of amateur athletes can be an important element in their health care and can help the athletes to achieve their full performance potential. Therefore, ergospirometric and hematological tests were performed. As a result, predictors for volume oxygen uptake were determined using a regression model, which included the following variables: the percentage of monocytes (*p* = 0.031), the concentration of sodium (*p* = 0.004), and total calcium (*p* = 0.03). All these parameters negatively affected VO2 max. Biochemical and physiological monitoring of amateur athletes can help to protect their health and prepare them properly for their training. The growing popularity of long-distance competitions among middle-aged amateur athletes and the lack of guidance on how to assess their health indicate the need for further research.

## 1. Introduction

Physical activity provokes adaptive changes in the body, which allow for a fuller use of physiological reserves. These changes are monitored, for example, in the circulatory-respiratory, nervous, and endocrine systems or body weight and composition. The maximum oxygen uptake (VO2 max) is known as the main determining factor for high performance potential (aerobic fitness). Research indicates that this is the main determinant of success in many disciplines, especially in endurance sports (e.g., cross-country skiing). VO2 max is genetically conditioned, and also influenced by body weight and composition, age, gender, diet and supplementation, and biochemical parameters of blood. Monitoring of these factors and the use of this knowledge can have a significant impact on endurance exercise capabilities and can help provide protection against overloading of the body and overtraining. This is particularly important in terms of participation in long-term endurance and strength efforts, such as long-distance races (40 km and more), and in the administration of effective training processes [1].

Adaptation changes also apply to amateur athletes, who are increasingly involved in sports competition. An example is the famous Vasaloppet race, with an attendance of more than 15,000 people. The Worldloppet series (20 races around the globe) attracted 64,000 skiers in the 2019/2020 season [2]. In addition to raising the capabilities of the amateur athletes, it is important to protect their health during long and demanding training sessions and races. This can be particularly relevant to men struggling with stress and around 40 years of age. They participate in sports competition often without prior, long-term, and adapted fitness preparation. This group is significantly vulnerable to the possibility of adverse health outcomes, such as early heart attacks. Above-average effort, like a ski marathon, contributes to this risk. However, the risk can be reduced by regular, fitted exercise that improves body composition and lipogram results and increases cardiovascular capacity and adaptation. Factors affecting the VO2 max level in professional athletes are well understood. Among amateur cross-country skiers, there is still a lack of information about the level of performance they present and the factors that significantly affect it [3]. This knowledge is essential for coaches to create an appropriate training plan for amateur athletes to not only improve their cardiovascular capacity but also support their health and protect against overtraining and its consequences. The aim of this research was to identify biochemical determinants for maximum oxygen uptake in amateur cross-country skiers. It is known that biochemical profiles are different between individuals who are characterized by inactive lifestyles and individuals who are professional athletes, however, the potential differences and similarities are poorly understood in individuals with distinctly elevated physical activity levels (e.g., long distance skiers), but who are not training professionally. In addition, the immune, endocrine, and hormonal systems’ responses to demanding physical exertion are poorly understood in this group. This knowledge can be important for doctors, trainers, and nutritionists to assess baseline levels 

## 2. Materials and Methods

### 2.1. Subjects

The study was conducted in accordance with the guidelines of Good Clinical Practice and the Helsinki Declaration. The study was approved by the Bioethics Committee at the Faculty of Human Nutrition and Consumption at Warsaw University of Life Sciences (SGGW) (No. 38p/2018, approved on 22 January 2019). All subjects gave their written consent before any testing. The study group consisted of 16 well-trained (but working full-time and living in a big city) male amateur cross-country skiers who had participated at least three times in long distance races (40 km or more) in the previous season. They had to give written consent before the test and were required to have a current medical certificate. The test was conducted in May at the end of transition period (i.e., the period after the last race and before the start of the preparation period). The goal of the research was to determine and evaluate the parameters of circulatory-respiratory fitness and predictors of maximum oxygen uptake.

### 2.2. Anthropometric Measurements

During the test, the weight and body composition of the participants were measured (using Tanita Body Composition Analyzer BODY IN MC-980 MA (Tokyo, Japan) consisting of an eight-point touch electrode system) just before the ergospirometry stress test began. The following were determined: body weight, water content, minerals, vitamins, fat content in the body (% and kg), lean muscle mass (muscle mass in % and kg), WHR (waist to hip ratio), and BMI (body mass index).

### 2.3. Measurement of Aerobic Capacity (VO2 max Test)

The ergospirometry stress test (time-to-exhaustion test) was conducted to assess the aerobic capacity of the participants and to determine their VO2 max (the level of maximum oxygen uptake). The treadmill HP Cosmos CPET equipment (Nussdorf-Traunstein, Germany) and Cosmed Quark/k4B2 (Rome, Italy) were used. The test started at a speed of 6 km/h and 0% inclination. Then, every 3 min the speed was increased by 1 km/h and the inclination by 1% until the subject reported the feeling of exhaustion. The heart rate of each participant was monitored using a Garmin ANT+ heart rate monitor (Olathe, KS, USA). This paper presents the maximum results of the test.

### 2.4. Venous Blood Sampling and Analysis

Venous blood was sampled from the subjects in the morning (7:00–10:00 AM), before the first meal, on the day of the exercise test. Alifax (Polverara, Italy) and the automatic methodology were set to measure ESR (erythrocyte sedimentation rate). Cobas 8000 (Basel, Switzerland) and spectrophotometric methods were used to measure creatinine, uric acid, urea, iron, magnesium, total calcium, amylase, alanine amino transferase (ALT), aspartate amino transferase (AST), gamma-glutamyl transpeptidase (GGTP, alkaline phosphatase (ALP), total bilirubin, and glucose. The spectrophotometric method was used to measure sodium and potassium, and the electrochemiluminescence immunoassay (ECLIA) method for the following hormone determinations: thyroid stimulating hormone (TSH), cortisol, and testosterone. The spectrophotometric method also measured lipid profile indicators (total cholesterol, high density lipoprotein-cholesterol (HDL-C), low density lipoprotein-cholesterol (LDL-C), and triglycerides), and C-reactive protein (CRP) was determined by immunoturbidimetric method.

### 2.5. Statistics

Variables were analyzed using the following basic descriptive statistics: number of persons (*N*), arithmetic mean, median, minimum (Min), maximum (Max), and standard deviation (*SD*). The Shapiro–Wilk’s test was used to evaluate the normality of the data. The Pearson correlation coefficients were used, whose values—in the case of statistical significance—can be interpreted as follows in Table 1. Regressive models were also developed for dependent variables, among which those with the highest determination factor *R*^2^ were selected. The value of 0.05 was assumed as the significance level (denoted by * *p* < 0.05, ** *p* < 0.01, *** *p* < 0.001).

## 3. Results

Detailed data characteristics of the study group and VO2 max are shown in Table 2.

### 3.1. Hematological Parameters of Participants

Hematological parameters of participants are shown in Table 3. All measured parameters, except bilirubin, were within the norm.

### 3.2. Correlations for Independent Variables

The VO2 max had significant correlations with five variables. Most of these correlations were moderately strong or strong, and positive. The results are shown in Table 4. Only statistically significant results are presented in the paper.

### 3.3. Regression Model

For the variable under consideration, a model with the highest *R*^2^^.^ coefficient value was selected, for which dependent variables could have influenced the variable. The distribution of the value of the *R*^2^ model adjustment measure depending on the selection of independent variables is presented in Figure 1.

## 4. Discussion

Studies have shown that athletes have their own inherent hematological and biochemical adaptations. It was also recorded that they are at a higher level compared to non-athletes in terms of physiological parameters [4]. This study investigated the differences in some hematological and biochemical parameters between amateurs, athletes, and non-athletes at rest. Consensus on the variability in hematological variables over time among athletes and non-athletes [5,6] or seasonal differences within the same squad [7] is lacking. This paper is, to our knowledge, the first providing information about amateur cross-country skiers.

A study by Baffour-Awuah et al. [8] showed differences between athletes and non-professional athletes. Comparing the results obtained in these studies, it can be noted that trained amateurs showed higher BMI parameters both compared to the training and non-training groups. According to Gallagher et al. [9], the body fat contents of the subjects were normal for age and gender, but comparing these results to athletes’ standards [10] shows higher levels of body fat (15.51 ± 2.59 vs. 10.5 ± 1.8) [11]. This could indicate the possibility of developing diseases associated with the cardiovascular system and, for example, overweight. However, the parameters of the patients’ lipid profile were correct. The body fat content should be related to a specific group that consists of amateur athletes. Exercise and regular training have a significant impact on lipid and lipoprotein levels in athletes. In fact, it has previously been documented that participation in sport has a positive effect on athletes compared to non-athletes with regard to lipid status markers [7].

The basic indicators defined in literature as significantly affecting the performance, such as the number of red blood cells, hemoglobin, and hematocrit were within the recommended standards, but were higher than in the study by Baffour-Awuah et al. This may be due to the selection of a group that only included men in this study. Studies in amateur cyclists [12] showed similar results of the red blood system at rest as in this study. Another study [13] suggested that some hematological values, such as the reticulocytes percentage (Ret%) and hemoglobin (Hb), were relatively stable over four consecutive seasons in elite triathletes, implying that in adults variability should be limited.

The results indicate that the immune system rates were lower than those recorded in the literature [7], but were normal. Their growth may indicate the body’s response to intense effort, pro-inflammatory factors, as well as immunosuppression resulting from prolonged fatigue. Studies have shown that hematological variables in athletes and non-athletes are subject to different influences after session or training [14].

Higher sodium and potassium values have been shown than in the study by Baffour-Awuah et al. [8], which may be due to lower training activities or other factors such as diet or environmental conditions. The correct concentrations of sodium and potassium determine the proper nerve conductivity and muscle tension.

Monitoring these indicators during the transition period (the period after the starts and before the preparation period in the annual training cycle for skiers takes place in April) can give guidelines in the field of diet, training and health care.

### 4.1. Predictors for VO2 max

The selected model includes the following variables: monocytes %, sodium, and total calcium (see Table 5). All of these parameters negatively affected the relative VO2 value (max) (i.e., as their value increases, the relative VO2 level (max) decreases).

#### 4.1.1. Monocytes

Monocytes are one of the largest types of blood cells. They make up about 3–7% of leukocytes. They are capable of reducing infectious conditions, as well as red blood cells and other large particles. However, they cannot replace the function of neutrophils in removing and destroying bacteria. Monocytes usually enter areas of the inflammatory tissue later than granulocytes. They are often found at places of chronic infection. They are precursors of the mononuclear macrophage system. After 1–2 days, they pass into the tissues, where they differentiate into macrophages. In addition to the role of scavengers, macrophages play a key role in immunity, taking antigens and processing them so that they can be recognized by foreign lymphocytes as foreign substances. They also release compounds that regulate the inflammatory process and produce interleukins, interferons, and leukotrienes. An increase in the level of monocytes is observed, for example, in bacterial, viral, and parasitic infections, autoimmune diseases at an early stage, cancer, and after intense physical exertion [15]. The importance of changes in monocyte properties in the systemic anti-inflammatory effect of exercise remains undetermined. Monocytes represent a relatively small part of all leukocytes.

This applies, in particular, to resting values in amateur athletes. In people who exercise regularly with moderate intensity, monocytes are less reactive to exogenous stimuli. The expression of TLR4 receptors and the percentage of monocytes with “inflammatory” CD14+/CD16+ after exercise are reduced, and the number of CD14+/CD16 monocytes at rest are also reduced [12]. The increased values of these cells can be affected by the stress that accompanies amateur athletes in everyday life and the result of a lack of adaptation to exercise and low capacity. The release of cortisol (especially after exercise) and catecholamines (during exercise) can stimulate the production of immune cells (in the first line of neutrophils and natural killers cells). This effect is differentiated by lymphocytes and monocytes having receptors for specific endocrine proteins and hormones. The nature of activity and gender determine the magnitude of these changes. Interval and strength training, involving, for example, fast-twitch fibers, can strongly stimulate these systems to work. In men, increased testosterone levels simultaneously affect immune functions through the macrophage system, lymphocytes, and muscle cells with adrenergic receptors, which can affect resting values [16].

According to other hypotheses, the high intensity of training with insufficient resting time will activate monocytes to produce pro-inflammatory cytokines, including IL-6 (Interleukin-6) and TNF-α (Tumor necrosis factor alpha) [17]. Subsequently, this can cause fatigue and negative changes in the immune system. That condition is characteristic for the starting period. High-intensity physical efforts that significantly disrupt homeostasis will contribute to this. At the same time, it can be assumed that the lack of adaptation to exercise can contribute to increased secretion of monocytes. This may be the case during a transitional period.

The results of this study may therefore indicate a weaker mobilization of the immune system during the introduction of cross-country skiers to training. Furthermore, the results of the regression indicate that reducing their number to reference values will have a positive impact on increasing exercise capacity. Changes in the body may result from the introduction of regular physical training after the recovery period or from a previous infection that often occurs during the spring solstice period. This may also be due to the lack of proper recovery. Monocytes can persist for quite a long time in patients’ blood after an infection. Most monocytes undergo apoptosis after 24 h. However, some of them may remain in the bloodstream. The half-life can then be up to 71 h [18]. This usually occurs when a patient in the course of the disease decreased the number of inert-absorbent granulocytes.

#### 4.1.2. Sodium

Sodium is responsible for regulating the water content in intercellular spaces. When there is potassium deficiency in the body, the amount of sodium (hypernatremia) increases excessively and the body retains water. Then there are swellings of the body. It can cause hypertension and heart disease, muscle spasms, mood swings. In this study, the value was normal, but these results exceeded those reported in the literature [8]. A favorable shift in the sodium/potassium balance of the diet in the general population may have a substantial impact on hypertension related diseases, including stroke and myocardial infarction [19]. Physical effort leads to increased losses of this mineral from the body along with sweat, which can quickly lead to disruption of water and electrolyte balance, muscle spasms, weakness and reduction of efficiency. Despite this, intake of extra sodium dose is not recommended for athletes. Isotonic drinks maintain hydration and normal mineral levels before, during and after exercise. This is especially true for winter disciplines and marathon runners, as confirmed by the latest research [20,21].

Many intend to consciously increase sodium intake in the days preceding and during competition, although these views appear informed mostly by nonscientific and/or non-evidence-based sources [22].

Sodium is often consumed by athletes during ultramarathons with the belief that sodium losses must be replaced to enhance performance and to prevent EAH, muscle cramping, and dehydration. This study shows that only adequate levels of sodium and potassium will guarantee high exercise capacity. Securing the supply of isotonic drinks during effort is enough. Increasing the amount of sodium in the daily supply will negatively affect the body and exercise capacity.

#### 4.1.3. Calcium

Calcium is needed for bone health, nerve conduction, and muscle excitation and contraction. It is a cofactor in glycogenolysis and works with vitamin K in blood coagulation and wound healing.

Although physical activity promotes bone density, athletes who already have low bone density and possibly longstanding suboptimal calcium intakes are likely to be at high risk of stress fractures when undertaking repetitive activities [23].

Previous studies confirmed the positive effect of calcium on muscle excitability, especially rapidly shrinking fibers. In this study, an increase in calcium levels results in lower exercise capabilities. Adverse effects of over-consumption include kidney disease, vascular calcification, increased risk of cardiovascular diseases, and impaired absorption of other minerals, such as magnesium, zinc, and iron [24]. Calcium also increases the incidence of heart spasms and an increase in its contractility, which increases blood pressure. The use of calcium channel blockers results in greater use of free fatty acids for the energy used in heart muscle contractions instead of glucose. This also translates into higher exercise potential [25]. However, the use of calcium channel inhibitors is prohibited in sports and included in the WADA (World Anti-Doping Agency) World List of Prohibited Substances [26]. Due to increased heart function (with an increase in heart rate and blood pressure), as well as the contraction of coronary vessels and coronary congestion, ischemia and myocardial infarctions may occur. The problem is compounded by too much activity and lack of adaptation to it. This underlines the role of regular and fitted training in amateur athletes (e.g., cross-country skiers) and people at risk of early heart attack [27].

Maintaining homeostasis is crucial. However, it is important to cause damage in the training process. The damage enables adaptation to occur. The supercompensation process is based on the capacity of chronic exercise to induce beneficial adaptive changes. However, it must be properly monitored so as not to lead to overtraining. Physical activity can induce beneficial adaptive changes, constitute a therapeutic tool in cardiovascular diseases, and cause antioxidant and anti-senescent effects in human cells [28].

## 5. Conclusions

The aim of this study was to identify determinants of the cardiovascular capacity of amateur long-distance skiers during the transition period. Three main predictors have been appointed for maximum oxygen uptake in this group. An increase in the percentage of monocytes, as well as changes in the concentration of sodium and calcium, can impair exercise capabilities in subjects and negatively affect their health. Monitoring these indicators can help to protect the health of amateur athletes and provide guidelines in the training process. This problem is described in detail in the case of professional athletes, but has not yet been studied among amateur athletes. Changes in hematological and biochemical parameters at rest need not be solely related to sports, indicating the need for similar research and special protection for men competing in ultramarathons and ski marathons. Biochemical and physiological monitoring of amateur athletes can help to protect their health and prepare them properly to the effort required to compete in their sport. The growing popularity of long-distance competitions among middle-aged amateur athletes and the lack of guidance on how to assess their health indicate the need for such research. Due to the relatively small sample size in this study, further research could focus on monitoring and assessing changes in other parts of the annual training cycle and in a larger group of athletes, including women.

## Figures and Tables

**Figure 1 diagnostics-10-00675-f001:**
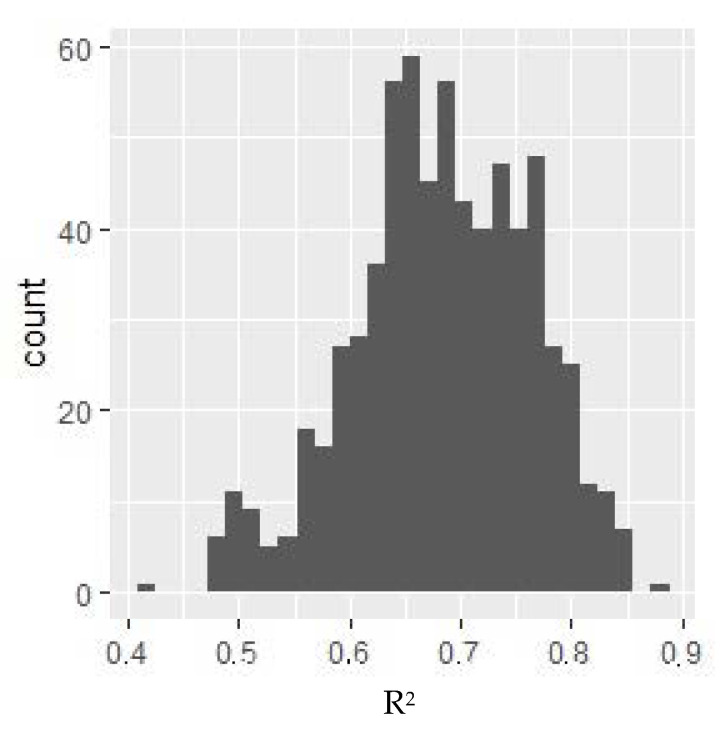
*R*^2^ value histogram for models for VO2 max. *Y*-axis: the number of models where *R*^2^ of the given value was obtained. The selected model includes the following variables: monocytes %, sodium, and total calcium. All of these parameters negatively affected the relative VO2 max (i.e., as their value increases, the relative VO2 max decreases).

**Table 1 diagnostics-10-00675-t001:** The Pearson correlation coefficients used in the case of statistical significance ^1^.

Coefficient (*r*)	Interpretation
0.0 ≤ |r|≤0.2	no correlation
0.2<|r|≤0.4	weak correlation
0.4<|r|≤0.7	average correlation
0.7<|r|≤0.9	strong correlation
0.9<|r|≤1.0	very strong correlation

^1^ The calculations were made in statistical software (ver. 3.6.0). (Chicaco, IL, USA).

**Table 2 diagnostics-10-00675-t002:** Anthropometric measurements and VO2 max level.

Variable	Arithmetic Average (*N* = 16) Means ± SD (Difference Δ—Delta)
Age (years)	38.69 ± 7.95 (28.00–56.00)
Body height (cm)	181.44 ± 6.53 (169.00–197.00)
Body mass (kg)	78.52 ± 6.18 (68.10–91.50)
Fat mass (kg)	12.22 ± 2.53 (7.90–16.00)
Fat mass (%)	15.51 ± 2.59 (10.00–19.30)
BMI (kg/m^2^)	23.84 ± 1.35 (21.00–25.70)
VO2 max (mL/kg/min)	48.37 ± 5.06 (38.54–55.81)

BMI: body mass index; *N*: number of patients; VO2 max: maximal oxygen uptake; SD: standard deviation. All data are presented as means ± standard deviation and the difference (Δ—delta).

**Table 3 diagnostics-10-00675-t003:** Hematological parameters of participants.

Morphology	Mean	Standard Deviation	Min	Max
Leukocytes (thou/µL)	5.4	0.88	4.2	9.1
Erythrocytes(M/µL)	5.06	0.29	4.2	6
Hemoglobin (g/dL)	15.19	0.5	14	18
Hematocrit %	43.86	1.53	40	51
Mean corpuscular value (MCV) (fL)	88.03	3.41	80	99
Mean corpuscular hemoglobin (MCH) (pg)	30.49	1.01	27	35
Mean corpuscular hemoglobin concentration (MCHC) (g/dL)	34.66	0.93	32	37
Platelets (thou/µL)	214.06	35.74	140	440
Red blood cell distribution width-standard deviation (RDW-SD) (fL)	40.77	2.97	35.1	43.9
Red blood cell distribution width-coefficient of variation (RDW-CV) %	12.98	0.79	11.6	14.4
Platelet distribution width (PDW) (fL)	13.45	2.14	9.8	16.1
Mean platelet volume (MPV) (fL)	10.64	1.06	9	13
Platelet-large cell ratio (P-LCR) %	31.83	8.69	13	43
Procalcitonin (PCT) %	0.21	0.04	0.2	0.4
Neutrophils (thou/µL)	2.7	0.55	2	7
Lymphocytes (thou/µL)	2.04	0.56	1	3.5
Monocytes (thou/µL)	0.45	0.09	0.2	1
Eosinophils (thou/µL)	0.25	0.19	0.1	0.5
Basophils (thou/µL)	0.03	0.03	0	0.1
Neutrophils %	49.69	7.91	40	70
Lymphocytes %	38.26	7.43	20	45
Eosinophils %	4.46	2.83	1	6
Basophils %	0.55	0.37	0	2
Erythrocyte sedimentation rate (ESR) (mm/h)	5.06	3.73	2	12
Urea (mg/dL)	33.94	6.43	10	50
Estimated glomerular filtration rate (eGFR) (mL/min/1.73m2)	73.03	13.4	-	-
Uric acid (mg/dL)	5.6	1.45	3.4	7
Glucose (mg/dL)	85.25	17.52	70	99
Total cholesterol (mg/dL)	179.1	32.58	115	190
Cholesterol high-density lipoproteins (HDL) (mg/dL)	58.21	12.17	≥45	-
Cholesterol non-HDL (mg/dL)	119.84	37.31	-	-
Cholesterol low-density lipoproteins (LDL) (mg/dL)	105.46	31.23	0	<115
Triglycerides (mg/dL)	81.36	34.51	0	150
Aspartate transaminase (AST) (U/L)	28.81	22.87	0	40
Alanine aminotransferase (ALT) (U/L)	22.22	7.59	0	41
Alkaline phosphatase (U/L)	59.22	10.44	40	129
Gamma-glutamyl transferase (GGTP) (U/L)	19.89	10.59	8	61
Serum amylase (U/L)	63.58	21.38	28	100
Sodium (mmol/L)	141.46	2.27	136	145
Potassium (mmol/L)	4.53	0.37	3.5	5.1
Total calcium (mmol/L)	2.42	0.11	2.15	2.5
Magnesium (mmol/L)	0.86	0.06	0.66	1.07
Iron (µg/dL)	11.05	50.3	33	193
C-reactive protein (CRP) (mg/dL)	0.71	0.97	0	5
Thyroid-stimulating hormone (µIU/mL)	1.71	0.71	0.27	4.2
Testosterone (ng/dL)	591.94	210.72	239	836
Cortisol (µg/dL) 7–10 AM	14.24	4.32	6.2	19.4

**Table 4 diagnostics-10-00675-t004:** The *p*-values and correlations for the VO2 max of the athletes.

Variable	*p*-Value	Correlation
Monocytes (thou/μL)	0.001	−0.750
Eosinophils (thou/μL)	0.026	0.613
Monocytes %	<0.001	−0.797
Eosinophils %	0.027	0.610
Erythrocyte sedimentation rate (ESR) (mm/h)	0.010	−0.620
Estimated glomerular filtration rate (eGFR) (mL/min/1.73 m^2^)	0.041	0.531
Sodium (mmol/L)	0.004	−0.680
Total calcium (mmol/L)	0.035	−0.530

**Table 5 diagnostics-10-00675-t005:** Multivariate linear regression model parameters for *r* VO2 max, *R*^2^ = 0.879.

Variable	Regression Coefficient	Statistical Error	*t*-Value	*p*-Value
Intercept	237.147	37.655	6.30	0.000
Monocytes %	−0.902	0.354	−2.55	0.031
Sodium (mmol/L)	−0.980	0.261	−3.76	0.004
Total Calcium (mmol/L)	−18.074	4.387	−4.12	0.03

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
