# Peer review of "Determinants of the Cardiovascular Capacity of Amateur Long-Distance Skiers during the Transition Period"

_diagnostics, 2020, doi:10.3390/diagnostics10090675_

Round 1

Reviewer 1 Report

In this work the author studied 16 well-trained amateur skiers. Anthropometric measurements, ergospirometry stress tests and blood investigations were performed on all participants.
The resultant VO2 was then correlated with each of the blood biomarkers and the most significantly correlated markers were selected for inclusion in a regression model.
The author concludes that in this small population of well-trained amateur skiers, monocyte count, calcium and sodium levels adversely impact the attained V02. Whilst it is plausible that these heamatological and biochemical factors could adversely affect the training capacity of athletes to warrant further in-depth analysis, I have a number of major concerns:

#The manuscript including the abstract need to be reviewed by a native English language speaker. Some typos need correcting (line 120, dupliate for which).
Some terms used are completely inappropriate 'the fat point' line 143.

#The abstract should include sample size for the cohort studied and some P-values should be reported for differences described.

#A key limitation is the absence of a control group and the small numbers of cases.

#In stats explain how you checked for normality of data.

#Table 3 showing a long list of all the blood test results is not particularly informative.

#The regression models are nonsensical firstly because of the incredibly small sample size but also because no attempt was made to adjust for basic things like sex, ethinicity, BMI etc. What method was used for the inclusion of variables (stepwise forward, hierarchal, backward, etc?). Were any of the imputed variables collinear?

Author Response

Dear Reviewer,

Thank you very much for your thorough analysis of my work and comments. I have responded to each of them. 

Kind regards.

Response to Reviewer 1 Comments

Thank you very much for your thorough analysis of my work and comments. I have responded to each of them.

 Point 1: The manuscript including the abstract need to be reviewed by a native English language speaker. Some typos need correcting (line 120, dupliate for which). 
Some terms used are completely inappropriate 'the fat point' line 143.

Response 1: The article was corrected. Thank you for your comments that identified significant errors.

Point 2:

#The abstract should include sample size for the cohort studied and some P-values should be reported for differences described.

Response 2: The changes were made. Information on the number of participants and statistical significance were added.

Point 3:

#A key limitation is the absence of a control group and the small numbers of cases.

Response 3: The research group included amateur skiers. The difficulty of testing athletes is in their small group, which is characterized by extraordinary training loads.  In the description of the research group, I mentioned the information of their daily physical activity.  In studies on the impact of maximum physical exertion and its biochemical monitoring, it is important to select people who will show results that are unusual for the population. There are big differences noted between inactive and active amateurs, as well as professional athletes. Therefore, the control group would not, in my view, be justified here. The results of the studies on the professional skiers are well described in the literature. However, the muscle damages and changes in the body in response to an extraordinary physical exertion in amateurs is still poorly studied. Only a few relate to changes in volume oxygen uptake and speed during the race (see for example: Carlsson, M.; Assarsson, H.; Carlsson, T. The influence of sex, age, and race experience on pacing profiles during the 90 km Vasaloppet ski race. J. Sports Med. 2016, 7, 11–19).  An additional difficulty is the lifestyle of these people. Physical activity is an addition to work and family life, but significantly exceeds the health-promoting recommendations of the World Health Organization. Therefore, it is difficult to compare this group to physically inactive people or professional athletes.

Point 4:

#In stats explain how you checked for normality of data.

Response 4: The Shapiro-Wilk's test was used. This information was added.

Point 5:

#Table 3 showing a long list of all the blood test results is not particularly informative.

Response 5:  Blood parameters were added to accurately described the results of amateur blood counts. I added them to show the characteristics of these people and that they did not deviate from general standards. In my opinion, this knowledge may indicate that not only deviations from the norm can negatively affect exercise capacity. That was the purpose of this, but of course the work could have been shortened by this table.

Point 6:

 The regression models are nonsensical firstly because of the incredibly small sample size but also because no attempt was made to adjust for basic things like sex, ethinicity, BMI etc. What method was used for the inclusion of variables (stepwise forward, hierarchal, backward, etc?). Were any of the imputed variables collinear?

Response 6 : It should be noticed that neither sex nor ethnicity varies in the analysed population (all subjects male caucasians) and the variety of BMI was rather small. Due to small sample size model it was decided that model should contain no more than 3 independent variables. Based on correlation coefficients analysis it was chosen which variables should be taken into account and every possible combination of three of them (as independent variables in model) was verified. Next, the combination with the highest R2 was presented as the best model.

Reviewer 2 Report

In this manuscript, the author would like to identify the determinants for the cardiovascular capacity among amateur long-distance skiers during the transition period. Finding these factors that affect of an amateur's heart can be a crucial element in these individuals' health care and also for them to attain their peak performance. Ergospirometric and hematological tests were conducted. Predictors for volume oxygen uptake were determined by applying a regression model. Changes in the percentage of monocytes, the concentrations of sodium and calcium were found to have the most significant negative impact on the cardiovascular capacity. Biochemical and physiological monitoring of amateurs could help to protect their health and prepare them appropriately to their sport activities.

(I) Major Comments

I have the following major comments.

(1) Page 2, lines 59-61,
The author stated that
"The study was approved by the Bioethics Committee at the Faculty of Human Nutrition and Consumption SGGW (No. 38p/2018). All subjects gave their written consent before any testing"

As shown in a previously published paper, Grzebisz N, Diagnostics (Basel). 2020;10:E442. PMID: 32629784, the above statement could be revised to
"The study was approved by the Bioethics Committee at the Faculty of Human Nutrition and Consumption at Warsaw University of Life Sciences (SGGW) (No. 38p/2018, approved on 22 January 2019). All subjects gave their written consent before any testing"

(2) Page 2, line 86,
The author stated that
"Alifax and the automatic methodology were set to measure OB"
It is unclear what the term "OB" refers to, because "OB" could refer to "obesity" or "leptin" or could have other meanings. Please clarify.

(3) Page 3, Table 3, first of all, this table lacks a table title. Please provide a pertinent title.
Second, there are a variety of non-English terms in the elements of the table, e.g., "Morfologia", "Średnia", "thou/µl", and "fl", and these should be converted to English terms (i.e., their English counterparts).
Third, the full names of table headers "SD", "L", and "H" shall be provided by the footnote.
Fourth, the full names of these terms in the table, i.e., "MCV", "MCH", "MCHC", "RDW-SD", "RDW-CV", "PDW", "MPV", "P-LCR", "PCT", "OB", "eGFR", "HDL", "LDL", "AST", "ALT", "GGTP", "CRP" and "TSG" shall be provided by the footnote.
Fifth, in the table's 1st column,
"Triglicerides" shall be corrected to "Triglycerides", and
it is what "Serum amalysis" is referring to, and could clarify.
Further, in Table 4, "thou/μl" should be converted to English term.
In Table 4's line 4,
For variable "Monocytes %", the P-value is shown as "0.000", which should be corrected to "<0.001",
and the Correlation is shown as "-0,797", which should be corrected to "0.797"

(4) Page 4, line 113, "VO2 variable relative (max) had significant correlations", and
Page 5, Table 4's title, "P-values and correlations for the VO2 variable relative (max) of the athletes"
In these texts, it is unclear what "VO2 variable relative (max)" refers to? Is this exactly same as "VO2(max)", or has a different definition? Please clarify.

Further, in Table 4, "thou/μl" should be converted to English term.
In Table 4's line 4,
For variable "Monocytes %", the P-value is shown as "0.000", which should be corrected to "<0.001",
and the Correlation is shown as "-0,797", which should be corrected to "0.797"
Also, in the table's 1st column, line 6,
for "eGFR" variable, why the unit is "ml/min/1.73m2" rather than "ml/min/m2"? Please double check and clarify, and also, the "2" in "m2" shall be in superscript.
In addition, the full names for "OB" and "eGFR" shall be provided by the footnote.

Furthermore, Page 4, lines 113-114, the author stated that
"The VO2 variable relative (max) had significant correlations with 8 variables. Most of these correlations were moderately strong or strong, and positive"
However, as shown in Table 4, among these 8 variables with significant correlations with "VO2 variable relative (max)", 5 variables, i.e., "Monocytes (thou/μl)", "Monocytes %", "OB", "eGFR", and "Total calcium",

(5) Page 5, lines 121-122, the author stated
"The distribution of the R2 model was shaped, depending on the selection of independent variables."
The meaning of the above statement is confusing, please clarity. Also, how is this statement related to Figure 1? Please elaborate.

(6) Page 5, line 124,
Figure 1's legend,
"R2 value histogram for models for relative VO2 dependent variable (max)"
It is unclear what is the definition of "relative VO2 dependent variable (max)"? Is this exactly same as "VO2(max)", or has a different definition? Please clarify.
Also, these following problems of Table 5 should be addressed.

In Table 5, 1st column, line 1, what does "free word" refer to? Please double-check and clarify.
In Table 5, line 2 and line 3, all the commas shall be dots for the numerical values, to keep a consistent format for numerical values.
In Table 5, "t-value" (4th column) and "p-value" (5th column), all the commas shall be dots for the numerical values, to keep a consistent format for numerical values.
In Table 5, 2nd column, column header "Factor", does this term refer to regression coefficient? If so, ""Factor" could be revised to "Regression coefficient".
In Table 5, there are these 4 dependent variables, and it is unclear whether the author created a multivariate linear regression model including all these 5 dependent variable altogether and the regression coefficients are then for this multivariate regression model,
or, the author created 5 different univariate linear regression models, including only a single dependent variable one by one, such that the regression coefficient for each dependent variable is that for the univariate regression model including only that single dependent variable? The author are suggested to clarify by adding a footnote.

(7) Page 5, Table 5, first of all, table's title,
"Regressive model parameters for relative VO2 variable(max), R2 = 0,879"
It is unclear what is the definition of "relative VO2 dependent variable (max)"? Is this exactly same as "VO2(max)", or has a different definition? Please clarify. Futher, in table's title, "R2 = 0,879" should be corrected to "R2 = 0.879"

Second, there are a variety of non-English terms in the elements of the table, e.g., "Morfologia", "Średnia", "thou/µl", and "fl", and these should be converted to English terms (i.e., their English counterparts).
Third, the full names of table headers "SD", "L", and "H" shall be provided by the footnote.
Fourth, the full names of these terms in the table, i.e., "MCV", "MCH", "MCHC", "RDW-SD", "RDW-CV", "PDW", "MPV", "P-LCR", "PCT", "OB", "eGFR", "HDL", "LDL", "AST", "ALT", "GGTP", "CRP" and "TSG" shall be provided by the footnote.

(8) Page 6, lines 162-163, the author stated
"The correct concentration of sodium and potassium affects the nervous phimitus and determines the proper nerve conductivity and muscle tension"
First of all,
"The correct concentration of sodium and potassium affects the nervous"
could be changed to
"The correct concentrations of sodium and potassium affect the nervous"
and
"determines the proper nerve conductivity"
could be changed to
"determine the proper nerve conductivity"
Second, it is unclear what "phimitus" refers to, and the author shall apply a commonly used term in the literature rather than this uncommon term.

(9) Page 6, line 171, the author stated
"Monocytes are one of the largest blood cells. They make up about 7 percent of leukocytes"
This statement is imprecise, and could be corrected to
"Monocytes are one of the largest type of blood cells. They make up about 3-7% of leukocytes (Zhou L, et al., Clin Vaccine Immunol. 2012;19:1065-74. PMID: 22552601)"

(10) Page 7, lines 185-187, the author stated
"The expression of TLR4 receptors and the percentage of monocytes with "inflammatory" CD14+/CD16+ phenonic after exercise is reduced and the number of CD14+/CD16 monocytes at rest is also reduced [11]"
First of all,
"is reduced and the number of CD14+/CD16 monocytes at rest is"
could be changed to
"are reduced and the numbers of CD14+/CD16 monocytes at rest are"
Second, it is unclear what "phenonic" refers to, and the author shall apply a commonly used term in the literature rather than this uncommon term.

(II) Minor Comments

There are a variety typographical and grammatical errors that should be corrected, and the author shall perform a thorough and careful proof-reading, and for example, these following errors should be corrected:

Page 1, line 16,
"the concentration of sodium and calcium have the most"
could be changed to
"the concentrations of sodium and calcium have the most"

Page 1, line 42,
"Above-average effort, like a ski marthon"
could be changed to
"Above-average effort, like a ski marathon"

Page 2, line 77,
"egospirometry stress test was used"
could be changed to
"ergospirometry stress test was used"

Page 2, line 88,
"total calcium, amylase, ALT, AST, GGTP, ALP, total bilirubin and glucose"
could be changed to
"total calcium, amylase, alanine amino transferase (ALT), aspartate amino transferase (AST), gamma-glutamyl transpeptidase (GGTP, alkaline phosphatase (ALP), total bilirubin and glucose"

Page 3, lines 89-90,
"and the ECLIA method for hormone determinations: TSH, cortisol and testosterone"
could be changed to
"and the electrochemiluminescence immunoassay (ECLIA) method for hormone determinations: thyroid stimulating hormone (TSH), cortisol and testosterone"

Page 3, line 91,
"indicators (cholesterol, HDL, LDL and triglycerides) and CRP was determined"
could be changed to
"indicators [total cholesterol, high density lipoprotein-cholesterol (HDL-C), low density lipoprotein-cholesterol (LDL-C), and triglycerides] and C-reactive protein (CRP) was determined"

Page 3, line 98,
"with the highest determination factor R2 were selected"
The "2" in "R2" shall be in superscript.

Page 3, Table 1,
"average corraletion"
could be changed to
"average correlation"

Page 3, Table 2,
"BMI (kg/m2)"
The "2" in "m2" shall be in superscript.

Page 3, line 108,
"Haematological parametres of participiants"
could be changed to
"Haematological parameters of participants"

Page 5, lines 120-121,
"for which for which dependent variables could have influenced the variable"
could be changed to
"for which dependent variables could have influenced the variable"

Page 5, lines 130-131,
"Studies have shown that athletes have its own inherent haematological and biochemical adaptations"
could be changed to
"Studies have shown that athletes have their own inherent haematological and biochemical adaptations"

Page 6, lines 139-141,
"higher BMI parameters both compared to the training and non-training group. According to Gallagher et al. [6] the body fat content of the subjects was normal for age and gender, but comparing these results to athletes [7] standards shows very high levels of body fat"
could be changed to
"higher BMI parameters both compared to the training and non-training groups. According to Gallagher et al. [6] the body fat contents of the subjects were normal for age and gender, but comparing these results to athletes' standards [7] shows very high levels of body fat"

Page 7, lines 193-194,
"Interval and strength training, involving e.g. fast-twich fibers, strongly stimulate"
could be changed to
"Interval and strength training, involving e.g. fast-twitch fibers, can strongly stimulate"

Author Response

Dear Reviewer,

Thank you very much for your thorough analysis of my work and comments. I have responded to each of them. 

Kind regards.

Response to Reviewer 2 Comments

 Point 1:

(1) Page 2, lines 59-61, 
The author stated that "The study was approved by the Bioethics Committee at the Faculty of Human Nutrition and Consumption SGGW (No. 38p/2018). All subjects gave their written consent before any testing"

As shown in a previously published paper, Grzebisz N, Diagnostics (Basel). 2020;10:E442. PMID: 32629784, the above statement could be revised to
"The study was approved by the Bioethics Committee at the Faculty of Human Nutrition and Consumption at Warsaw University of Life Sciences (SGGW) (No. 38p/2018, approved on 22 January 2019). All subjects gave their written consent before any testing"

Response 1:

The statement was revised to give more information about the approval of the Committee.

Point 2:

(2) Page 2, line 86,
The author stated that
"Alifax and the automatic methodology were set to measure OB"
It is unclear what the term "OB" refers to, because "OB" could refer to "obesity" or "leptin" or could have other meanings. Please clarify.

Response 2:

The acronym OB. was translated to ESR and explained (erythrocyte sedimentation rate).

Point 3:

(3) Page 3, Table 3, first of all, this table lacks a table title. Please provide a pertinent title.
Second, there are a variety of non-English terms in the elements of the table, e.g., "Morfologia", "Średnia", "thou/µl", and "fl", and these should be converted to English terms (i.e., their English counterparts).
Third, the full names of table headers "SD", "L", and "H" shall be provided by the footnote.
Fourth, the full names of these terms in the table, i.e., "MCV", "MCH", "MCHC", "RDW-SD", "RDW-CV", "PDW", "MPV", "P-LCR", "PCT", "OB", "eGFR", "HDL", "LDL", "AST", "ALT", "GGTP", "CRP" and "TSG" shall be provided by the footnote.
Fifth, in the table's 1st column, 
"Triglicerides" shall be corrected to "Triglycerides", and it is what "Serum amalysis" is referring to, and could clarify.
Further, in Table 4, "thou/μl" should be converted to English term.
In Table 4's line 4,
For variable "Monocytes %", the P-value is shown as "0.000", which should be corrected to "<0.001",
and the Correlation is shown as "-0,797", which should be corrected to "0.797"

Response 3:

A table title was added. The full names were put onto the table, and the units were converted to English terms.

Point 4:

(4) Page 4, line 113, "VO2 variable relative (max) had significant correlations", and 
Page 5, Table 4's title, "P-values and correlations for the VO2 variable relative (max) of the athletes"
In these texts, it is unclear what "VO2 variable relative (max)" refers to? Is this exactly same as "VO2(max)", or has a different definition? Please clarify.

Further, in Table 4, "thou/μl" should be converted to English term.
In Table 4's line 4,
For variable "Monocytes %", the P-value is shown as "0.000", which should be corrected to "<0.001",
and the Correlation is shown as "-0,797", which should be corrected to "0.797"
Also, in the table's 1st column, line 6,
for "eGFR" variable, why the unit is "ml/min/1.73m2" rather than "ml/min/m2"? Please double check and clarify, and also, the "2" in "m2" shall be in superscript.
In addition, the full names for "OB" and "eGFR" shall be provided by the footnote.

Furthermore, Page 4, lines 113-114, the author stated that
"The VO2 variable relative (max) had significant correlations with 8 variables. Most of these correlations were moderately strong or strong, and positive"
However, as shown in Table 4, among these 8 variables with significant correlations with "VO2 variable relative (max)", 5 variables, i.e., "Monocytes (thou/μl)", "Monocytes %", "OB", "eGFR", and "Total calcium",

Response 4:

Corrected. VO2max is exactly the same as VO2 variable relative. The unit for eGFR variable is ml/min/1.73m2 because of the correction for body surface. I changed the number of variables. I treated e.g. monocytes in % and nominal terms as two variables.

Point 5:

(5) Page 5, lines 121-122, the author stated
"The distribution of the R2 model was shaped, depending on the selection of independent variables."
The meaning of the above statement is confusing, please clarity. Also, how is this statement related to Figure 1? Please elaborate.

Response 5:  

The distribution of the value of the R2 model was presented on Figure 1.

Point 6:

Page 5, line 124,
Figure 1's legend, 
"R2 value histogram for models for relative VO2 dependent variable (max)"
It is unclear what is the definition of "relative VO2 dependent variable (max)"? Is this exactly same as "VO2(max)", or has a different definition? Please clarify.
Also, these following problems of Table 5 should be addressed.

In Table 5, 1st column, line 1, what does "free word" refer to? Please double-check and clarify.
In Table 5, line 2 and line 3, all the commas shall be dots for the numerical values, to keep a consistent format for numerical values.
In Table 5, "t-value" (4th column) and "p-value" (5th column), all the commas shall be dots for the numerical values, to keep a consistent format for numerical values.
In Table 5, 2nd column, column header "Factor", does this term refer to regression coefficient? If so, ""Factor" could be revised to "Regression coefficient".
In Table 5, there are these 4 dependent variables, and it is unclear whether the author created a multivariate linear regression model including all these 5 dependent variable altogether and the regression coefficients are then for this multivariate regression model,
or, the author created 5 different univariate linear regression models, including only a single dependent variable one by one, such that the regression coefficient for each dependent variable is that for the univariate regression model including only that single dependent variable? The author are suggested to clarify by adding a footnote

Response 6:

VO2max is exactly the same as relative VO2 dependent variable (max). I standardized it at work. "Free word" should be translated as "intercept". It was a mistake. There should be a footnote under the table that it was a multivariate linear regression model. I added this information.

Point 7:

 Page 5, Table 5, first of all, table's title,
"Regressive model parameters for relative VO2 variable(max), R2 = 0,879"
It is unclear what is the definition of "relative VO2 dependent variable (max)"? Is this exactly same as "VO2(max)", or has a different definition? Please clarify. Futher, in table's title, "R2 = 0,879" should be corrected to "R2 = 0.879"

Second, there are a variety of non-English terms in the elements of the table, e.g., "Morfologia", "Średnia", "thou/µl", and "fl", and these should be converted to English terms (i.e., their English counterparts).
Third, the full names of table headers "SD", "L", and "H" shall be provided by the footnote.
Fourth, the full names of these terms in the table, i.e., "MCV", "MCH", "MCHC", "RDW-SD", "RDW-CV", "PDW", "MPV", "P-LCR", "PCT", "OB", "eGFR", "HDL", "LDL", "AST", "ALT", "GGTP", "CRP" and "TSG" shall be provided by the footnote.

Response 7:

Please refer to response 3 and 6.

Point 8:

(8) Page 6, lines 162-163, the author stated
"The correct concentration of sodium and potassium affects the nervous phimitus and determines the proper nerve conductivity and muscle tension"
First of all, 
"The correct concentration of sodium and potassium affects the nervous"
could be changed to
"The correct concentrations of sodium and potassium affect the nervous"
and 
"determines the proper nerve conductivity"
could be changed to
"determine the proper nerve conductivity"
Second, it is unclear what "phimitus" refers to, and the author shall apply a commonly used term in the literature rather than this uncommon term.

Response 8:

Corrected.

Point 9:

 (9) Page 6, line 171, the author stated
"Monocytes are one of the largest blood cells. They make up about 7 percent of leukocytes"
This statement is imprecise, and could be corrected to
"Monocytes are one of the largest type of blood cells. They make up about 3-7% of leukocytes (Zhou L, et al., Clin Vaccine Immunol. 2012;19:1065-74. PMID: 22552601)"

Response 9:

Corrected.

Point 10:

(10) Page 7, lines 185-187, the author stated
"The expression of TLR4 receptors and the percentage of monocytes with "inflammatory" CD14+/CD16+ phenonic after exercise is reduced and the number of CD14+/CD16 monocytes at rest is also reduced [11]"
First of all, 
"is reduced and the number of CD14+/CD16 monocytes at rest is"
could be changed to
"are reduced and the numbers of CD14+/CD16 monocytes at rest are"
Second, it is unclear what "phenonic" refers to, and the author shall apply a commonly used term in the literature rather than this uncommon term.

Response 10:

Corrected. ,,Phenonic’’ was deleted as unclear and useless.

 (II) Minor Comments

Response: Corrected.

Reviewer 3 Report

The article is original and well written. The methodology is interesting. The results are well documented and discussed. The theme is interesting, the statistical analysis is well done, the methods are clear and replicable. The figures and tables are clear and eligible, the conclusions correlate to the results found. The paper is easy to follow and logical.  

Author Response

Thank you for the review. 

Kind regards,

Natalia Grzebisz

Reviewer 4 Report

This study deals with an important topic that is the influence of type, volume and intensity of exercise training in determining athletes’ health.

It is clear and well-written.

I have a couple of concerns:

The most important of them regards the lack of discussion on the capacity of chronic exercise (if well monitored) to induce beneficial adaptive changes, therefore to represent a therapeutic approach.

In my opinion, this is a very important issue (now just mentioned) that, instead, it is necessary to deepen in the manuscript.

There are several studies that could be useful (e.g. The anti-ageing molecule sirt1 mediates beneficial effects of cardiac rehabilitation. Immun Ageing. 2017 Mar 16;14:7.; Cardiovascular Effects and Benefits of Exercise. Front Cardiovasc Med. 2018 Sep 28;5:135.).

Other points:

-Table 3: please, translate from polish!

- it is necessary to indicate what is the sex of the enrolled subjects.

Author Response

Thank you very much for your thorough analysis of my work and comments. I have responded to each of them.

Response to Reviewer 4 Comments

Point 1:

The most important of them regards the lack of discussion on the capacity of chronic exercise (if well monitored) to induce beneficial adaptive changes, therefore to represent a therapeutic approach.

In my opinion, this is a very important issue (now just mentioned) that, instead, it is necessary to deepen in the manuscript.

There are several studies that could be useful (e.g. The anti-ageing molecule sirt1 mediates beneficial effects of cardiac rehabilitation. Immun Ageing. 2017 Mar 16;14:7.; Cardiovascular Effects and Benefits of Exercise. Front Cardiovasc Med. 2018 Sep 28;5:135.).

Response 1: Thank you for your comments. Information about beneficial adaptive changes  is enriching. I added a paragraph on the role of destruction in the process of adaptation and development in the discussion.

Point 2: Table 3: please, translate from polish!

Response 2: Translated.

Point 3: It is necessary to indicate what is the sex of the enrolled subjects

Response 3:  I added this information.

Reviewer 5 Report

Diagnostics; diagnostics-900348

Title: Determinants of the Cardiovascular Capacity of Amateur Long-Distance Skiers during the Transition Period

Grzebisz examined anthropometric, physiological, and biochemical markers per levels of maximum oxygen uptake in middle-aged male amateur long-distance skiers. In brief, the author found that there were significant predictors of maximum oxygen uptake as percentage of monocytes and changes in sodium and calcium concentrations. This information could be useful for understanding general cardiovascular health before and during athletic training in groups susceptible to attacks of myocardial ischemia and stroke. See comments below.

(1) The primary concern with the manuscript is that it is not clear how this study significantly differs from a previously published one that perhaps used the same/similar cohort (PMID: 32629784). In such manner, it is not understood why the incrementally new findings in the current manuscript were not reported in that same study. Further, the study could be strengthened with having age- and sex-matched sedentary individuals and/or professional athletes as controls. Finally, the presentation of the manuscript is casual and should include more thorough citation (e.g., no references in the Introduction) and significantly less descriptions (e.g., “very important”, “quite a long time”) in lieu of statements that provide quantitative information typically accompanied by numerical values and statistics. For contrast, the author’s most recent publication (PMID: 32629784) is more appropriately presented as a science study and paper.      

(2) Key information is missing such as the use of males only in the Materials and Methods, definition and duration of the “transition period”, meaning of the y-axis in Figure 1 (“count” of what and how is this informative?), and definitions of some abbreviations (e.g., “OB”, “EAH”) as examples.

Author Response

Dear Reviewer,

Thank you very much for your thorough analysis of my work and comments. I have responded to each of them. 

Kind regards.

Response to Reviewer 5 Comments

 Point 1: The primary concern with the manuscript is that it is not clear how this study significantly differs from a previously published one that perhaps used the same/similar cohort (PMID: 32629784). In such manner, it is not understood why the incrementally new findings in the current manuscript were not reported in that same study. Further, the study could be strengthened with having age- and sex-matched sedentary individuals and/or professional athletes as controls. Finally, the presentation of the manuscript is casual and should include more thorough citation (e.g., no references in the Introduction) and significantly less descriptions (e.g., “very important”, “quite a long time”) in lieu of statements that provide quantitative information typically accompanied by numerical values and statistics. For contrast, the author’s most recent publication (PMID: 32629784) is more appropriately presented as a science study and paper.  

Response 1:

Thank you very much for your comments. The aim of the study was to identify biochemical indicators and increase knowledge of laboratory diagnostics in this area. The purpose of separating these results was to draw attention to these two groups of predictors, which could have a significant impact on the performance and health of the subjects. Combining them into a single work would limit the possibility of extensive discussion and evaluation of the results. In my opinion, their separation can be better used by biochemists or physiologists as a concrete reference to their specialisation.  Hence the decision to separate the articles.

The control group would be the most enriching factor in the study. However, there is still a lack of results from studies on biochemical changes, particularly endocrine and immunological changes, in the amateur group. However, they differ from non actve person or competitive athletes. Private amateur’s time limits the ability to unify a group. These are some of the many aspects and problems that arise when monitoring athletes. In the future, it would certainly be developing for science to broaden the methodology of research.

All of that, I added references in the introduction and corrected the wording.

Point 2: Key information is missing such as the use of males only in the Materials and Methods, definition and duration of the “transition period”, meaning of the y-axis in Figure 1 (“count” of what and how is this informative?), and definitions of some abbreviations (e.g., “OB”, “EAH”) as examples.

Response 2:

Thank you for your valuable comments. I added information about the period studied, definitions of some abbreviations. Y axis presents the number of models where R2 of the given value was obtained.

Round 2

Reviewer 1 Report

The salient points appear to have been addressed. There are still some typos in the abstract and main manuscript so further independent review of the manuscript by a native English language reader would still be recommended. Due to the small sample size consider toning down some of the conclusions. 

Author Response

To tone done conclusions i added information:

Due to relatively small sample size further research could focus on monitoring and assessing changes in other parts of the annual training cycle and on a larger group of athletes, including women. These are important public health issues.

The abstract was reviewed. 

Reviewer 5 Report

The manuscript has been improved. My remaining suggestions include defining "very high levels of body fat" in Line 143 & "quite a long time" as Lines 212-213.  

Author Response

Precise information on body fat mass in elite cross country skies and monocyte half-life were added (including citations).